# Watch-And-Help: A Challenge for Social Perception and Human-AI Collaboration

**Xavier Puig**[1]   **Tianmin Shu**[1]   **Shuang Li**[1]   **Zilin Wang**[2]   **Yuan-Hong Liao**[3,5]
**Joshua B. Tenenbaum**[1]   **Sanja Fidler**[3,4,5]   **Antonio Torralba**[1]

[1]Massachusetts Institute of Technology   [2]ETH Zurich
[3]University of Toronto   [4]NVIDIA   [5]Vector Institute

## Abstract

In this paper, we introduce Watch-And-Help (WAH), a challenge for testing social intelligence in agents. In WAH, an AI agent needs to help a human-like agent perform a complex household task efficiently. To succeed, the AI agent needs to i) understand the underlying goal of the task by watching a single demonstration of the human-like agent performing the same task (social perception), and ii) coordinate with the human-like agent to solve the task in an unseen environment as fast as possible (human-AI collaboration). For this challenge, we build VirtualHome-Social, a multi-agent household environment, and provide a benchmark including both planning and learning based baselines. We evaluate the performance of AI agents with the human-like agent as well as with real humans using objective metrics and subjective user ratings. Experimental results demonstrate that the proposed challenge and virtual environment enable a systematic evaluation on the important aspects of machine social intelligence at scale.[1]

## 1 Introduction

Humans exhibit altruistic behaviors at an early age (Warneken & Tomasello, 2006). Without much prior experience, children can robustly recognize goals of other people by simply watching them act in an environment, and are able to come up with plans to help them, even in novel scenarios. In contrast, the most advanced AI systems to date still struggle with such basic social skills.

In order to achieve the level of social intelligence required to effectively help humans, an AI agent should acquire two key abilities: i) social perception, i.e., the ability to understand human behavior, and ii) collaborative planning, i.e., the ability to reason about the physical environment and plan its actions to coordinate with humans. In this paper, we are interested in developing AI agents with these two abilities.

Towards this goal, we introduce a new AI challenge, Watch-And-Help (WAH), which focuses on social perception and human-AI collaboration. In this challenge, an AI agent needs to collaborate with a human-like agent to enable it to achieve the goal faster. In particular, we present a 2-stage framework as shown in Figure 1. In the first, *Watch* stage, an AI agent (Bob) watches a human-like agent (Alice) performing a task once and infers Alice's goal from her actions. In the second, *Help* stage, Bob helps Alice achieve the same goal in a different environment as quickly as possible (i.e., with the minimum number of environment steps).

This 2-stage framework poses unique challenges for human-AI collaboration. Unlike prior work which provides a common goal a priori or considers a small goal space (Goodrich & Schultz, 2007; Carroll et al., 2019), our AI agent has to reason about what the human-like agent is trying to achieve by watching a single demonstration. Furthermore, the AI agent has to generalize its acquired knowl-

---

[1]Code and documentation for the VirtualHome-Social environment are available at https://virtual-home.org. Code and data for the WAH challenge are available at https://github.com/xavierpuigf/watch_and_help. A supplementary video can be viewed at https://youtu.be/lrB4K2i8xPI.

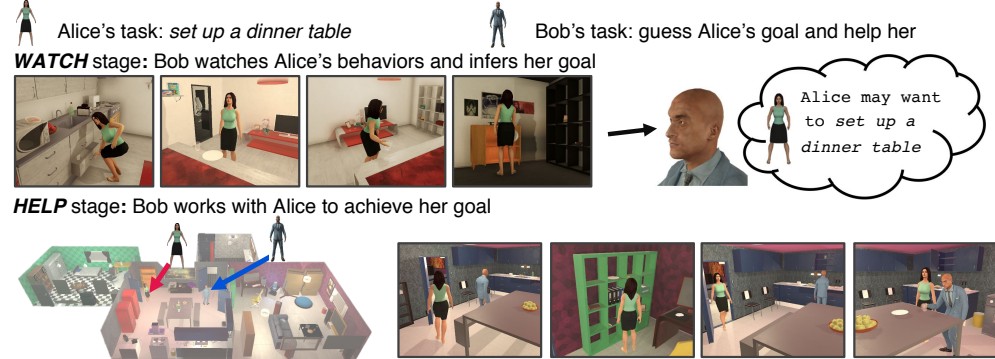

Figure 1: Overview of the Watch-And-Help challenge. The challenge has two stages: i) in the *Watch* stage, Bob will watch a single demonstration of Alice performing a task and infer her goal; ii) then in the *Help* stage, based on the inferred goal, Bob will work with Alice to help finish the same task as fast as possible in a *different* environment.

edge about the human-like agent's goal to a new environment in the *Help* stage. Prior work does not investigate such generalization.

To enable multi-agent interactions in realistic environments, we extend an open source virtual platform, VirtualHome (Puig et al., 2018), and build a multi-agent virtual environment, VirtualHome-Social. VirtualHome-Social simulates realistic and rich home environments where agents can interact with different objects (e.g, by opening a container or grabbing an object) and with other agents (e.g., following, helping, avoiding collisions) to perform complex tasks. VirtualHome-Social also provides i) built-in agents that emulate human behaviors, allowing training and testing of AI agents alongside virtual humans, and ii) an interface for human players, allowing evaluation with real humans and collecting/displaying human activities in realistic environments (a functionality key to machine social intelligence tasks but not offered by existing multi-agent platforms). We plan to open source our environment.

We design an evaluation protocol and provide a benchmark for the challenge, including a goal inference model for the *Watch* stage, and multiple planning and deep reinforcement learning (DRL) baselines for the *Help* stage. Experimental results indicate that to achieve success in the proposed challenge, AI agents must acquire strong social perception and generalizable helping strategies. These fundamental aspects of machine social intelligence have been shown to be key to human-AI collaboration in prior work (Grosz & Kraus, 1996; Albrecht & Stone, 2018). In this work, we demonstrate how we can systematically evaluate them in more realistic settings at scale.

The main contributions of our work are: i) a new social intelligence challenge, Watch-And-Help, for evaluating AI agents' social perception and their ability to collaborate with other agents, ii) a multi-agent platform allowing AI agents to perform complex household tasks by interacting with objects and with built-in agents or real humans, and iii) a benchmark consisting of multiple planning and learning based approaches which highlights important aspects of machine social intelligence.

## 2 RELATED WORK

**Human activity understanding.** An important part of the challenge is to understand human activities. Prior work on activity recognition has been mostly focused on recognizing short actions (Sigurdsson et al., 2018; Caba Heilbron et al., 2015; Fouhey et al., 2018), predicting pedestrian trajectories (Kitani et al., 2012; Alahi et al., 2016), recognizing group activities (Shu et al., 2015; Choi & Savarese, 2013; Ibrahim et al., 2016), and recognizing plans (Kautz, 1991; Ramırez & Geffner, 2009). We are interested in the kinds of activity understanding that require inferring other people's mental states (e.g., intentions, desires, beliefs) from observing their behaviors. Therefore, the *Watch* stage of our challenge focuses on the understanding of humans' goals in a long sequence of actions instead. This is closely related to work on computational Theory of Mind that aims at inferring

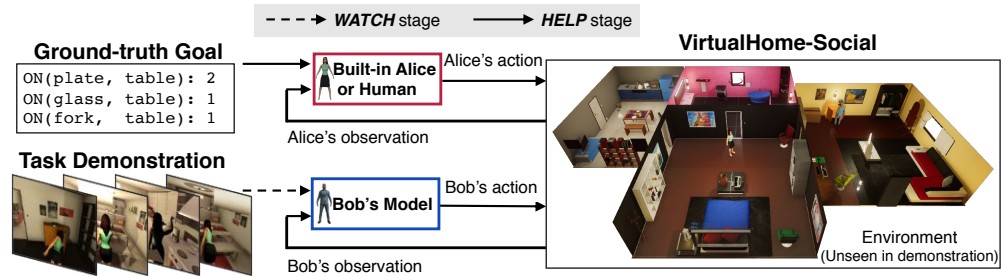

Figure 2: The system setup for the WAH challenge. An AI agent (Bob) watches a demonstration of a human-like agent (Alice) performing a task, and infers the goal (a set of predicates) that Alice was trying to achieve. Afterwards, the AI agent is asked to work together with Alice to achieve the same goal in a new environment as fast as possible. To do that, Bob needs to plan its actions based on i) its understanding of Alice's goal, and ii) a partial observation of the environment. It also needs to adapt to Alice's plan. We simulate environment dynamics and provide observations for both agents in our VirtualHome-Social multi-agent platform. The platform includes a built-in agent as Alice which is able to plan its actions based on the ground-truth goal, and can react to any world state change caused by Bob through re-planning at every step based on its latest observation. Our system also offers an interface for real humans to control Alice and work with an AI agent in the challenge.

humans' goals by observing their actions (Baker et al., 2017; Ullman et al., 2009; Rabinowitz et al., 2018; Shum et al., 2019). However, in prior work, activities were simulated in toy environments (e.g., 2D grid worlds). In contrast, this work provides a testbed for conducting Theory-of-Mind type of activity understanding in simulated real-world environments.

**Human-robot interaction.** The helping aspect of the WAH challenge has been extensively studied in human-robot interaction (HRI). However, prior work in HRI has been mainly restricted in lab environments (Goodrich & Schultz, 2007; Dautenhahn, 2007; Nikolaidis et al., 2015; Rozo et al., 2016), and the goals in the collaborative tasks were either shared by both agents or were defined in a small space. The setup in WAH is much more challenging – the goal is sampled from a large space, needs to be inferred from a single demonstration, and must be performed in realistic and diverse household environments through a long sequence of actions.

**Multi-agent virtual environments**. There has been a large body of platforms for various multi-agent tasks (Jaderberg et al., 2019; Samvelyan et al., 2019; OpenAI, 2018; Lowe et al., 2017; Resnick et al., 2018; Shu & Tian, 2018; Carroll et al., 2019; Suarez et al., 2019; Baker et al., 2019; Bard et al., 2020). However, these multi-agent platforms can only simulate simple or game-like environments and do not support for human-AI collaborations on real-life activities. Existing platforms for realistic virtual environments mainly focus on single agent settings for tasks such as navigation (Savva et al., 2019; Xia et al., 2018; Brodeur et al., 2017; Zhu et al., 2017; Xia et al., 2018) , embodied question answering (Gordon et al., 2017; Wijmans et al., 2019; Das et al., 2018), or single agent task completion (Puig et al., 2018; Shridhar et al., 2019; Misra et al., 2018; Gao et al., 2019). In contrast, the proposed VirtualHome-Social environment allows AI agents to engage in multi-agent household activities by i) simulating realistic and interactive home environments, ii) incorporating humanoid agents with human-like behaviors into the system, iii) providing a wide range of commands and animations for navigation and object manipulation, and iv) allowing human participation. Because of these features, VirtualHome-Social can serve as a testbed for complex social perception and human-AI collaboration tasks, which is complementary to existing virtual environments.

## 3 THE WATCH-AND-HELP CHALLENGE

The Watch-And-Help challenge aims to study AI agents' ability to help humans in household activities. To do that, we design a set of tasks defined by predicates describing the final state of the environment. For each task, we first provide Bob a video that shows Alice successfully performing the activity (*Watch* stage), and then place both agents in a new environment where Bob has to help Alice achieve the same goal with the minimum number of time steps (*Help* stage).

Figure 2 provides an overview of the system setup for the Watch-And-Help challenge. For this challenge, we build a multi-agent platform, VirtualHome-Social (Section 4), that i) supports concurrent actions from multiple agents and ii) provides observations for the agents. Alice represents a built-in agent in the system; she plans her actions based on her own goal and a partial observation of the environment. Bob serves as an external AI agent, who does not know Alice's ground-truth goal and only has access to a single demonstration of Alice performing the same task in the past. During the *Help* stage, Bob receives his observation from the system at each step and sends an action command back to control the avatar in the environment. Alice, on her part, updates her plan at each step based on her latest observation to reflect any world state change caused by Bob. We also allow a human to control Alice in our system. We discuss how the system and the built-in agent work in Section 4.

**Problem Setup.** Formally, each task in the challenge is defined by Alice's goal $g$ (i.e., a set of goal predicates), a demonstration of Alice taking actions to achieve that goal $D = \{s_{\text{Alice}}^t, a_{\text{Alice}}^t\}_{t=1}^T$ (i.e., a sequence of states $s_{\text{Alice}}^t$ and actions $a_{\text{Alice}}^t$), and a new environment where Bob collaborates with Alice and help achieve the same goal as quickly as possible. During training, the ground-truth goal of Alice is shown to Bob as supervision; during testing, Bob no longer has access to the ground-truth goal and thus has to infer it from the given demonstration.

**Goal Definitions.** We define the goal of a task as a set of predicates and their counts, which describes the target state. Each goal has 2 - 8 predicates. For instance, "ON(plate, dinnertable):2; ON(wineglass, dinnertable):1" means "putting two plates and one wine glass onto the dinner table." The objects in a predicate refer to object classes rather than instances, meaning that any object of a specified class is acceptable. This goal definition reflects different preferences of agents (when setting up a dinner table, some prefer to put water glasses, others may prefer to put wine glasses), increasing the diversity in tasks. We design five predicate sets representing five types of household activities: 1) setting up a dinner table, 2) putting groceries / leftovers to the fridge, 3) preparing a simple meal, 4) washing dishes, and 5) reading a book while having snacks or drinks. In total, there are 30 different types of predicates. In each task, the predicates of a goal are sampled from one of the five predicate sets (as a single household activity). More details about the predicate sets and goal definitions are listed in Appendix B.1.

## 4 VIRTUALHOME-SOCIAL

Building machine social intelligence for real-life activities poses additional challenges compared to typical multi-agent settings, such as far more unconstrained goal and action spaces, and the need to display human actions realistically for social perception.

With that in mind, we create VirtualHome-Social, a new environment where multiple agents (including real humans) can execute actions concurrently and observe each other's behaviors. Furthermore, we embed planning-based agents in the environment as virtual humans that AI agents can reason about and interact with.

In the rest of this section, we describe the observations, actions, and the built-in human-like agent provided in VirtualHome-Social. Appendix A includes more information.

**Observation space**. The environment supports symbolic and visual observations, allowing agents to learn helping behaviors under different conditions. The symbolic observations consist on a scene graph, with nodes representing objects and edges describing spatial relationships between them.

**Action space**. Agents can navigate in the environment and interact with objects in it. To interact with objects, agents need to specify an action and the index of the intended object (e.g., "grab $\langle 3 \rangle$" stands for grabbing the object with id 3). An agent can only interact with objects that are within its field of sight, and therefore its action space changes at every step.

**Human-like agents.** To enable a training and testing environment for human-AI interactions, it is critical to incorporate built-in agents that emulate humans when engaging in multi-agent activities. Carroll et al. (2019) has attempted to train policies imitating human demonstrations. But those policies would not reliably perform complex tasks in partially observable environments. Therefore, we devise a planning-based agent with bounded rationality, provided as part of the platform. This agent operates on the symbolic representation of its partial observation of the environment. As shown in Figure 3, it relies on two key components: 1) a belief of object locations in the environment

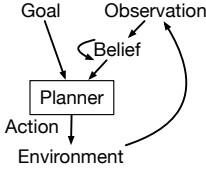

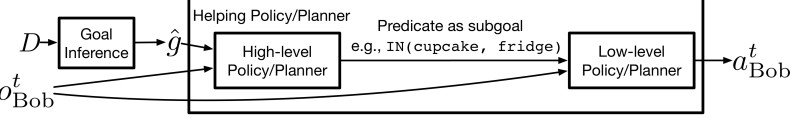

Figure 3: Overview of the human-like agent.

Figure 4: The overall design of the baseline models. A goal inference model infers the goal from a demonstration $D$ and feeds it to a helping policy (for learning-based baselines) or to a planner to generate Bob's action. We adopt a hierarchical approach for all baselines.

(Figure 13 in Appendix A.3), and 2) a hierarchical planner, which uses Monte Carlo Tree Search (MCTS) (Browne et al., 2012) and regression planning (RP) (Korf, 1987) to find a plan for a given goal based on its belief. At every step, the human-like agent updates its belief based on the latest observation, finds a new plan, and executes the first action of the plan concurrently with other agents. The proposed design allows agents to robustly perform tasks in partially observable environments while producing human-like behaviors[2]. We provide more details of this agent in Appendix A.3.

## 5 BENCHMARK

### 5.1 EVALUATION PROTOCOL

**Training and Testing Setup.** We create a training set with 1011 tasks and 2 testing sets (test-1, test-2). Each test set has 100 tasks. We make sure that i) the helping environment in each task is different from the environment in the pairing demonstration (we sample a different apartment and randomize the initial state), and ii) goals (predicate combinations) in the test set are unseen during training. To evaluate generalization, we also hold out 2 apartments for the *Help* stage in the test sets. For the training set and test-1 set, all predicates in each goal are from the same predicate set, whereas a goal in test-2 consists of predicates sampled from two different predicates sets representing multi-activity scenarios (e.g., putting groceries to the fridge and washing dishes). Note that during testing, the ground-truth goals are not shown to the evaluated Bob agent. More details can be found in Appendix B. An episode is terminated once all predicates in Alice's goal are satisfied (i.e., a success) or the time limit (250 steps) is reached (i.e., a failure).

**Evaluation Metrics.** We evaluate the performance of an AI agent by three types of metrics: i) success rate, ii) speedup, and iii) a cumulative reward. For speedup, we compare the episode length when Alice and Bob are working together ($L_{\text{Help}}$) with the episode length when Alice is working alone ($L_{\text{Alice}}$), i.e., $L_{\text{Alice}}/L_{\text{Bob}} - 1$. To account for both the success rate and the speedup, we define the cumulative reward of an episode with $T$ steps as $R = \sum_{t=1}^{T} \mathbb{1}(s^t = s_g) - 0.004$, where $s^t$ is the state at step $t$, $s_g$ is the goal state. $R$ ranges from -1 (failure) to 1 (achieving the goal in zero steps).

### 5.2 BASELINES

To address this challenge, we propose a set of baselines that consist of two components as shown in Figure 4: a goal inference model and a goal-conditioned helping planner / policy. In this paper, we assume that the AI agent has access to the ground-truth states of objects within its field of view (but one could also use raw pixels as input). We describe our approach for the two components below.

**Goal inference.** We train a goal inference model based on the symbolic representation of states in the demonstration. At each step, we first encode the state using a Transformer (Vaswani et al., 2017) over visible objects and feed the encoded state into a long short-term memory (LSTM) (Hochreiter & Schmidhuber, 1997). We use average pooling to aggregate the latent states from the LSTM over time and build a classifier for each predicate to infer its count. Effectively, we build 30 classifiers, corresponding to the 30 predicates in our taxonomy and the fact that each can appear multiple times.

---

[2]We conducted a user study rating how realistic were the trajectories of the agents and those created by humans, and found no significant difference between the two groups. More details can be found in Appendix D.4.

**Helping policy/planner.** Due to the nature of the tasks in our challenge – e.g., partial observability, a large action space, sparse rewards, strict preconditions for actions – it is difficult to search for a helping plan or learn a helping policy directly over the agent's actions. To mitigate these difficulties, we propose a hierarchical architecture with two modules for both planning and RL-based approaches as shown in Figure 4. At every step, given the goal inferred from the demonstration, $\hat{g}$, and the current observation of Bob, a high-level policy or planner will output a predicate as the best subgoal to pursue for the current step; the subgoal is subsequently fed to a low-level policy or planner which will yield Bob's action $a_{\text{Bob}}^t$ at this step. In our baselines, we use either a learned policy or a planner for each module. We use the symbolic representation of visible objects as Bob's observation $o_{\text{Bob}}^t$ for all models. We summarize the overall design of the baseline models as follows (please refer to Appendix C for the details of models and training procedures):

**HP**: A hierarchical planner, where the high-level planner and the low-level planner are implemented by MCTS and regression planning (RP) respectively. This is the same planner as the one for Alice, except that i) it has its own partial observation and thus a different belief from Alice, and ii) when given the ground-truth goal, the high-level planner uses Alice's plan to avoid overlapping with her.

**Hybrid**: A hybrid model of RL and planning, where an RL policy serves as the high-level policy and an RP is deployed to generated plans for each subgoal sampled from the RL-based high-level policy. This is to train an agent equipped with basic skills for achieving subgoals to help Alice through RL.

**HRL**: A hierarchical RL baseline where high-level and low-level policies are all learned.

**Random**: A naive agent that takes a random action at each step.

To show the upper bound performance in the challenge, we also provide two oracles:

**Oracle$^{\text{B}}$**: An HP-based Bob agent with full knowledge of the environment and the true goal of Alice.

**Oracle$^{\text{A, B}}$**: Alice has full knowledge of the environment too.

## 5.3 RESULTS

We evaluate the *Watch* stage by measuring the recognition performance of the predicates. The proposed model achieves a precision and recall of 0.85 and 0.96 over the test-1 set. To evaluate the importance of seeing the full demonstration, we test a model that takes as input the graph representation of the last observation, leading to a precision and recall of 0.79 and 0.75. When using actions taken by Alice as the input, the performance increases to a precision and recall of 0.99 and 0.99. The chance precision and recall is 0.08 and 0.09.

We report the performance of our proposed baselines (average and standard error across all episodes) in the *Help* stage in Figure 5. In addition to the full challenge setup, we also report the performance of the helping agents using true goals (indicated by the subscript $_{\text{TG}}$) and using random goals (by $_{\text{RG}}$), and the performance of Alice working alone. Results show that planning-based approaches are the most effective in helping Alice. Specifically, **HP$_{\text{TG}}$** achieves the best performance among non-oracle baselines by using the true goals and reasoning about Alice's future plan, avoiding redundant actions and collisions with her (Figure 6 illustrates an example of collaboration). Using the inferred goals, both **HP** and **Hybrid** can offer effective help. However, with a random goal inference (**HP$_{\text{RG}}$**), a capable Bob agent becomes counter productive – frequently undoing what Alice has achieved due to their conflicting goals (conflicts appear in 40% of the overall episodes, 65% for *Put Groceries* and *Set Meal*). This calls for an AI agent with the ability to adjust its goal inference dynamically by observing Alice's behavior in the new environment (e.g., Alice correcting a mistake made by Bob signals incorrect goal inference). **HRL** works no better than **Random**, even though it shares the same global policy with **Hybrid**. While the high level policy selects reasonable predicates to perform the task, the low level policy does not manage to achieve the desired goal. In most of the cases, this is due to the agent picking the right object, but failing to put it to the target location afterwards. This suggests that it is crucial for Bob to develop robust abilities to achieve the subgoals. There is no significant difference between **Random** and **Alice** baselines ($t(99) = -1.38, p = 0.17$).

We also evaluate the baselines in the test-2 set, containing tasks with multiple activities. The goal inference model achieves a precision and recall of 0.68 and 0.64. The performance gap from test-1 indicates that the model fails to generalize to generalize to multi-activity scenarios, overfitting to predicate combinations seen during training. For the *Help* stage, we evaluate the performance of

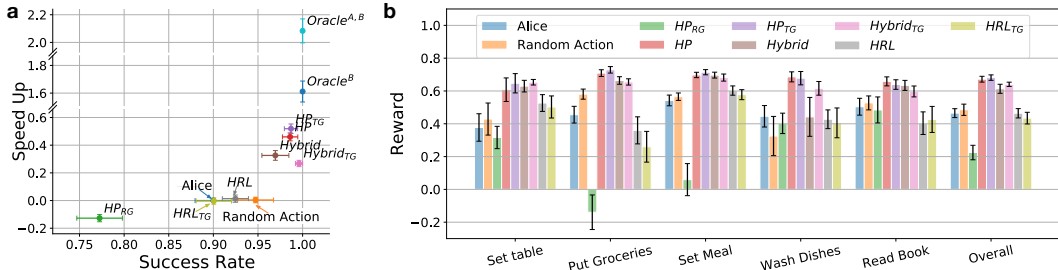

Figure 5: a) Success rate (x axis) and speedup (y axis) of all baselines and oracles. The performance of an effective Bob agent should fall into the upper-right side of the Alice-alone baseline in this plot. b) Cumulative reward in the overall test set and in each household activity category (corresponding to the five predicate sets introduced in Section 3).

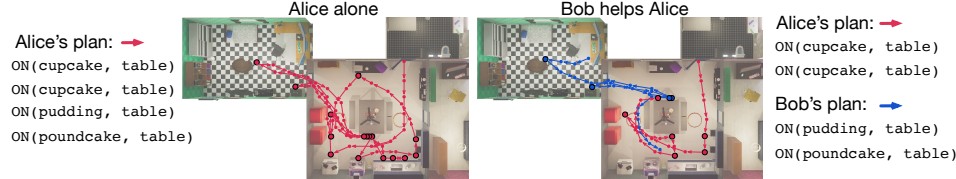

Figure 6: Example helping plan. The arrows indicate moving directions and the circles with black borders indicate moments when agents interacted with objects. When working alone (left), Alice had to search different rooms; but with Bob's help (right), Alice could finish the task much faster.

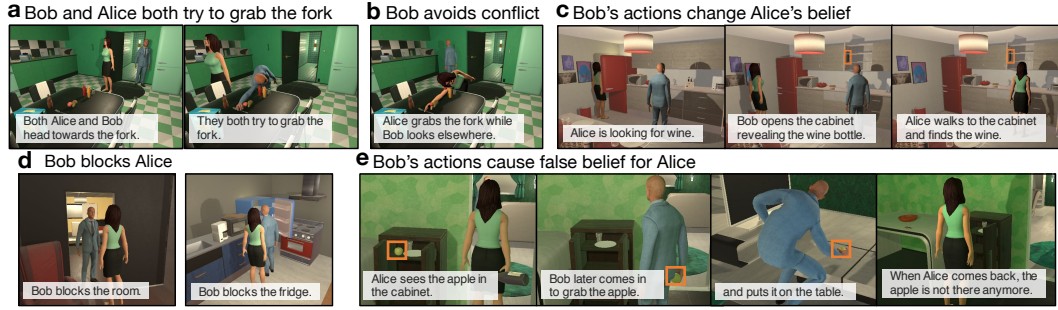

Figure 7: Example helping behaviors. We show more examples in the supplementary video.

Alice alone, as well as the best performing baseline, **HP**. Alice achieves a success rate of $95.40 \pm 0.01$, while the **HP** baseline achieves a success rate of $88.60 \pm 0.02$ and a speedup of $0.21 \pm 0.04$. Compared to its performance in the test-1 set, the **HP** baseline suffers a significant performance degradation in the test-2 set, which is a result of the lower goal recognition accuracy in the *Watch* stage.

To better understand the important factors for the effectiveness of helping, we analyze the helping behaviors exhibited in our experiments and how they affect Alice from the following aspects.

**Predicting Alice's Future Action.** When coordinating with Alice, Bob should be able to predict Alice's future actions to efficiently distribute the work and avoid conflicts (Figure 7ab).

**Helping Alice's Belief's Update.** In addition to directly achieving predicates in Alice's goal, Bob can also help by influencing Alice's belief update. A typical behavior is that when Bob opens containers, Alice can update her belief accordingly and find the goal object more quickly (Figure 7c). This is the main reason why Bob with random actions can sometimes help speed up the task too.

**Multi-level Actions.** The current baselines do not consider plans over low-level actions (e.g., pathfinding). This strategy significantly decreases the search space, but will also result in inefficient

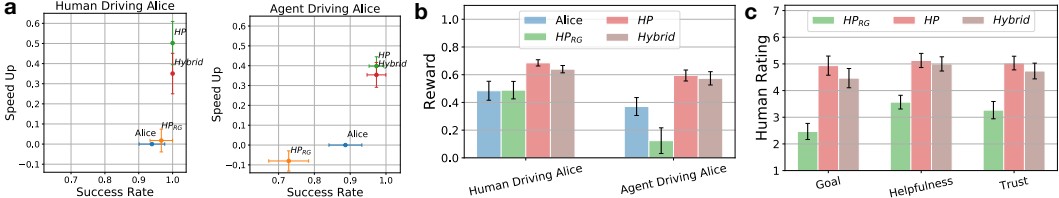

Figure 8: a) Success rate (x axis) and speedup (y axis). b) Cumulative reward with real humans or with the human-like agent. c) Subjective ratings from Exp. 2. Here, **Alice** refers to humans or the human-like agent acting alone, whereas **HP**, **Hybrid**, and **HP**$_{RG}$ indicate different AI agents helping either humans or the human-like agent. All results are based on the same 30 tasks in the test set.

pathfinding and inability to predict other agents' future paths. Consequently, Bob agent sometimes unintentionally blocks Alice (Figure 7d). A better AI agent should consider actions on both levels.

**False Belief.** Actions taken by an agent may cause another agent to have false beliefs (Figure 7e).

## 6 HUMAN EXPERIMENTS

Our ultimate goal is to build AI agents that can work with real humans. Thus, we further conduct the following two human experiments, where Alice is controlled by a real human.

**Experiment 1: Human performing tasks alone.** In this experiment, we recruited 6 subjects to perform tasks alone by controlling Alice. Subjects were given the same observation and action space as what the human-like agent had access to. They could click one of the visible objects (including all rooms) and select a corresponding action (e.g., "walking towards", "open") from a menu to perform. They could also choose to move forward or turn left/right by pressing arrow keys. We evaluated 30 tasks in the test set. Each task was performed by 2 subjects, and we used the average steps they took as the single-agent performance for that task, which is then used for computing the speedup when AI agents help humans. The performance of a single agent when being controlled by a human or by a human-like agent in these 30 tasks is shown in Fig. 8ab with the label of **Alice**. Human players are slightly more efficient than the human-like agent but the difference is not significant, as reported by the t-test over the number of steps they took ($t(29) = -1.63, p = .11$).

**Experiment 2: Collaboration with real humans.** This experiment evaluates how helpful AI agents are when working with real humans. We recruited 12 subjects and conducted 90 trials of human-AI collaboration using the same 30 tasks as in Exp. 1. In each trial, a subject was randomly paired with one of three baseline agents, **HP**, **Hybrid**, and **HP**$_{RG}$, to perform a task. After each trial, subjects were asked to rate the AI agent they just worked with on a scale of 1 to 7 based on three criteria commonly used in prior work (Hoffman, 2019): i) how much the agent knew about the true goal (1 - no knowledge, 4 - some knowledge, 7 - perfect knowledge), ii) how helpful you found the agent was (1 - hurting, 4 - neutral, 7 - very helpful), and iii) whether you would trust the agent to do its job (1 - no trust, 4 - neutral, 7 - full trust). For a fair comparison, we made sure that the random goal predictions for **HP**$_{RG}$ were the same as the ones used in the evaluation with the human-like agent.

As shown Figure 8, the ranking of the three baseline AI agents remains the same when the human-like agent is replaced by real humans, and the perceived performance (subjective ratings) is consistent with the objective scores. We found no significant difference in the objective metrics between helping humans and helping the human-like agent; the only exception is that, when paired with real humans, **HP**$_{RG}$ had a higher success rate (and consequently a higher average cumulative reward). This is because humans recognized that the AI agent might have conflicting subgoals and would finish other subgoals first instead of competing over the conflicting ones with the AI agent forever, whereas the human-like agent was unable to do so. Appendix D.3 shows an example. This adaption gave humans a better chance to complete the full goal within the time limit. We provide more details of the procedures, results, and analyses of the human experiments in Appendix D.

## 7 CONCLUSION

In this work, we proposed an AI challenge to demonstrate social perception and human-AI collaboration in common household activities. We developed a multi-agent virtual environment to test an AI agent's ability to reason about other agents' mental states and help them in unfamiliar scenarios. Our experimental results demonstrate that the proposed challenge can systematically evaluate key aspects of social intelligence at scale. We also show that our human-like agent behaves similarly to real humans in the proposed tasks and the objects metrics are consistent with subject ratings.

Our platform opens up exciting directions of future work, such as online goal inference and direct communication between agents. We hope that the proposed challenge and virtual environment can promote future research on building more sophisticated machine social intelligence.

ACKNOWLEDGMENTS

The information provided in this document is derived from an effort sponsored by the Defense Advanced Research Projects Agency (DARPA), and awarded to Raytheon BBN Technologies under Contract Number HR001120C0022.

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

# A  VIRTUALHOME-SOCIAL

## A.1  COMPARISON WITH EXISTING PLATFORMS

There have been many virtual environments designed for single-agent and multi-agent tasks. Table 1 summarizes the key features of the proposed VirtualHome-Social in comparison with existing virtual platforms. The key features of our environment include i) multiple camera views, ii) both high-level and low-level actions, iii) humanoid avatars with realistic motion simulations, iv) built-in human-like agents emulating human behaviors in household activities, and v) multi-agent capacities.

Critically, VirtualHome-Social enables collecting and displaying human activities in realistic environments, which is a key function necessarily for social perception and human-AI collaboration. In contrast, existing multi-agent platforms do no offer such functionality.

Table 1: We compare VirtualHome-Social with existing embodied single-agent and multi-agent platforms on the following aspects: 1) action space (high-level actions and/or low-level actions), 2) views (3rd person and/or egocentric views), 3) realistic environments, 4) humanoid agents, 5) human-like built-in agents that other agents can interact with, and 6) multi-agent capabilities.

| Platform | Action | Views | Realistic | Humanoid | Human-like Agent | Multi-agent |
|---|---|---|---|---|---|---|
| Overcooked (Carroll et al., 2019) | High/Low | 3rd Person | No | No | Yes | Yes |
| Malmo (Johnson et al., 2016) | High/Low | 3rd Person/Ego | No | No | No | Yes |
| ThreeDWorld (Gan et al., 2020) | High/Low | 3rd Person/Ego | Yes | No | No | Yes |
| VRKitchen (Gao et al., 2019) | High/Low | 3rd Person/Ego | Yes | Yes | No | No |
| AI2-THOR (Kolve et al., 2017) | High/Low | Ego | Yes | No | No | Yes |
| House3D (Wu et al., 2018) | Low | Ego | Yes | No | No | No |
| HoME (Brodeur et al., 2017) | Low | Ego | Yes | No | No | No |
| Gibson (Xia et al., 2018) | Low | Ego | Yes | No | No | No |
| AI Habitat (Savva et al., 2019) | Low | Ego | Yes | No | No | No |
| VirtualHome-Social | High/Low | 3rd Person/Ego | Yes | Yes | Yes | Yes |

## A.2  ENVIRONMENT DESCRIPTION

The environment is composed of different apartments with objects that can be placed to generate diverse scenes for the *Watch* and *Help* stages. Each object contains a class name, a set of states, 3D coordinates and an index for identification, which is needed for action commands that involve object interaction. The object indices are unique and consistent in the scene so that an agent can track the identities of individual objects throughout an episode.

### A.2.1  APARTMENTS

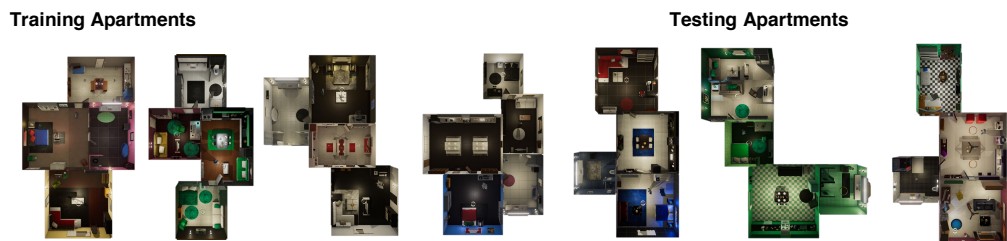

Figure 9: Apartments used in VirtualHome-Social. The last two apartments are uniquely used as helping environments during the testing phase.

We provide 7 distinctive apartments in total as shown in Figure 9. For the purpose of testing agents' generalization abilities, in the Watch-And-Help challenge, the last two apartments are held out for the helping environments in the testing set exclusively.

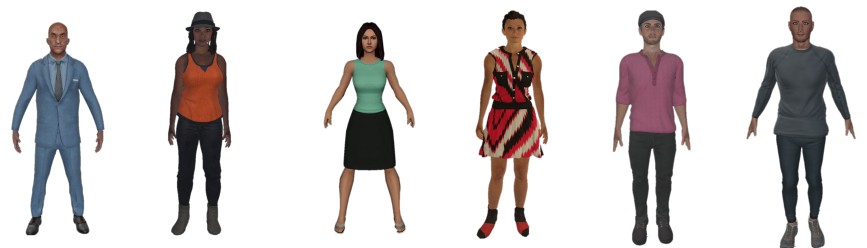

Figure 10: Avatars available in VirtualHome-Social.

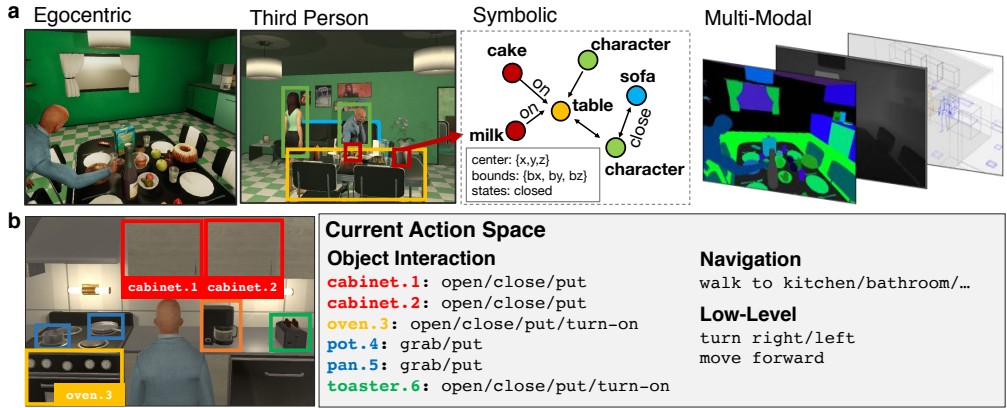

Figure 11: a) VirtualHome-Social provides egocentric views, third-person views and scene graphs with symbolic state representations of objects and agents. It also offers multi-modal inputs (RGB, segmentation, depth, 3D boxes and skeletons). b) Illustration of the action space at one step.

### A.2.2 AVATARS

VirtualHome-Social provides a pool of diverse humanoid avatars (see Figure 10). This allows us to randomly sample different avatars for both agents in the Watch-And-Help challenge. We hope this can help reduce the biases in the environment. The supplementary video shows an example of this, where the clothing color indicates the role of each agent. For the public release of the platform, we intend to further increase the diversity of the avatar pool.

### A.2.3 OBSERVATION

The environment supports symbolic and visual observations (Figure 11a), allowing agents to learn helping behaviors under different conditions. The visual observations provide RGB, depth, semantic and instance segmentation, albedo and luminance, normal maps, 3D skeletons and bounding boxes. Building upon Liao et al. (2019), we represent the symbolic observations as a state graph with each node representing the class label and physical state of an object, and each edge representing the spatial relation of two objects. The environment also provides multiple views and supports both full observability and partial observability settings.

We show examples of the observations in the supplementary video. In addition to the world states, our system also allows users to include direct messages from other agents as part of the observation for an agent.

### A.2.4 ACTION SPACE

As shown in Figure 11b, agents in VirtualHome-Social can perform both high-level actions, such as navigating towards a known location, or interacting with an observed object, and low-level actions,

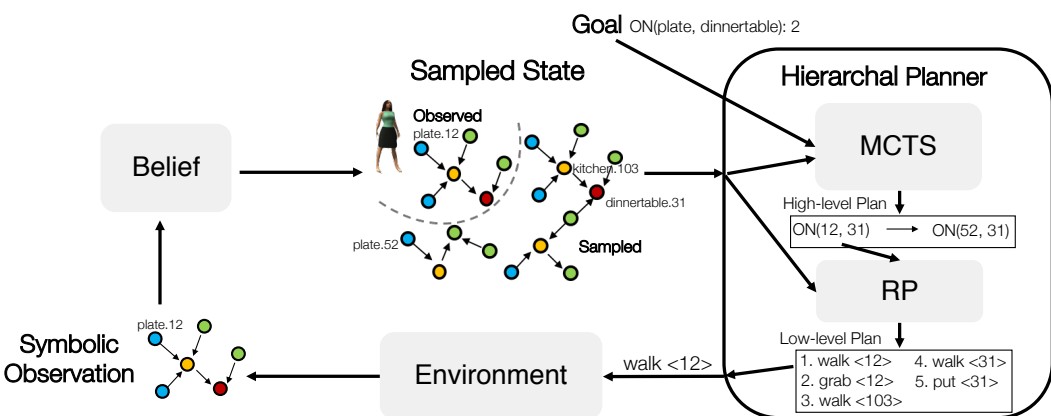

Figure 12: Schematic of the human-like agent. Based on the state graph sampled from the belief, the hierarchical planner searches for a high-level plan over subgoals using MCTS; then RP searches for a low-level plan over actions for each subgoal. The first action of each plan is sent back to the environment for execution.



Figure 13: The agent's belief is represented as the location distribution of objects, and is updated at each step based on the previous belief and the latest observation. In the example, the open cabinet reveals that the wine glass can not be in there, and that there is an apple inside, updating the belief accordingly.

such as turning or moving forward for a small step. For actions involving interactions with entities (objects or other agents), an agent needs to specify the indices of the intended entities (e.g., "grab $\langle 3 \rangle$" stands for grabbing the object with id 3). An agent can only interact with objects that are within its field of sight, and therefore its action space changes at every step. When executing navigation actions, an agent can only move 1 meter towards the target location within one step. On average, an agent's action space includes 167 different actions per step.

## A.3 HUMAN-LIKE AGENT

We discuss how the human-like agent works in more details here. The agent pipeline can be seen in Figure 12. The agent has access to a partial observation of the environment, limited to the objects that are in the same room and not in some closed container. The agent is equipped with a belief module (Figure 13), that gives information about the unseen objects, under the assumption that the existence of objects in the environment is known, but not their location. For each object in the environment, the belief contains a distribution of the possible locations where it could be. We adopt uniform distributions as the initial belief when the agent has not observed anything.

At each time, the agent obtains a partial observation, and updates its belief distribution accordingly. Then, the belief module samples a possible world state from the current distribution. To ensure that the belief state is consistent between steps, we only resample object locations that violate the current belief (e.g. an object was believed to be in the fridge but the agent sees that the fridge is in fact empty).

Based on the sampled state, a hierarchical planner will search for the optimal plan for reaching the goal, based on the goal definition. Specifically, we use MCTS to search for a sequence of subgoals

(i.e., predicates), and then each subgoal is fed to a regression planner (RP) that will search for an action sequence to achieve the subgoal. For the high-level planner, the subgoal space is obtained by the intersection between what predicates remained to be achieved and what predicates could be achieved based on the sampled state. Note here each subgoal would specify an object instance instead of only the object class defined in the goal so that the low-level planner will be informed which object instances it needs to interact with. For instance, in the example illustrated in Figure 12, there are two plates (whose indices are 12, 52) and the dinner table's index is 31 according to the sampled state. There are two unsatisfied goal predicates (i.e., two `ON(plate, dinnertable)`), then a possible subgoal space for the high-level planner would be $\{$`ON(12, 31)`, `ON(52, 31)`$\}$. For RP, it starts from the state defined by the subgoal and searches for the low-level plan backward until it finds an action that is part of the current action space of the agent.

To mimic human behaviors in a home setting, we also expect the human-like agent to close containers unless it needs to look inside or put objects into them. For that, we augment the MCTS-based high-level planner with heuristics for the closing behavior – the agent will close an container when it finds no relevant goal objects inside or has already grabbed/put in the all target objects out of that container. We find that this augmentation makes the overall agent behaviors closer to what a real human would do in a household environment.

Thanks to the hierarchical design, the planner for the human-like agent can be run in real-time (on average, replanning at each step only takes 0.05 second). This also gives the agent a bounded rationality, in that the plan is not the most optimal but is reasonably efficient. The optimality of the planner can be further tuned by the hyper-parameters of MCTS, such as the number of simulation, the maximum number steps in the rollouts, and the exploration coefficients.

### A.4 Specifications

The environment can be run in a single or multiple processes. A single process runs at 10 actions per second. We train our models using 10 processes in parallel.

## B More Details on the Challenge Setup

### B.1 Predicate Sets for Goal Definitions

Table 2: Predicate sets used for defining the goal of Alice in five types of activities.

| | |
|---|---|
| Set up a dinner table | `ON(plate,dinnertable)`,`ON(fork,dinnertable)`, `ON(waterglass,dinnertable)`,`ON(wineglass,dinnertable)` |
| Put groceries | `IN(cupcake,fridge)`,`IN(pancake,fridge)`,`IN(poundcake,fridge)`, `IN(pudding,fridge)`,`IN(apple,fridge)`, `IN(juice,fridge)`,`IN(wine,fridge)` |
| Prepare a meal | `ON(coffeepot,dinnertable)`,`ON(cupcake,dinnertable)`, `ON(pancake,dinnertable)`,`ON(poundcake,dinnertable)`, `ON(pudding,dinnertable)`,`ON(apple,dinnertable)`, `ON(juice,dinnertable)`,`ON(wine,dinnertable)` |
| Wash dishes | `IN(plate,dishwasher)`,`IN(fork,dishwasher)`, `IN(waterglass,dishwasher)`,`IN(wineglass,dishwasher)` |
| Read a book | `HOLD(Alice,book)`,`SIT(Alice,sofa)`,`ON(cupcake,coffeetable)`, `ON(pudding,coffeetable)`,`ON(apple,coffeetable)`, `ON(juice,coffeetable)`,`ON(wine,coffeetable)` |

Table 2 summarizes the five predicate sets used for defining goals. Note that VirtualHome-Social supports more predicates for potential future extensions on the goal definitions.

### B.2 Training and Testing Setup

During training, we randomly sample one of the 1011 training tasks for setting up a training episode. For evaluating an AI agent on the testing set, we run each testing task for five times using different random seeds and report the average performance.

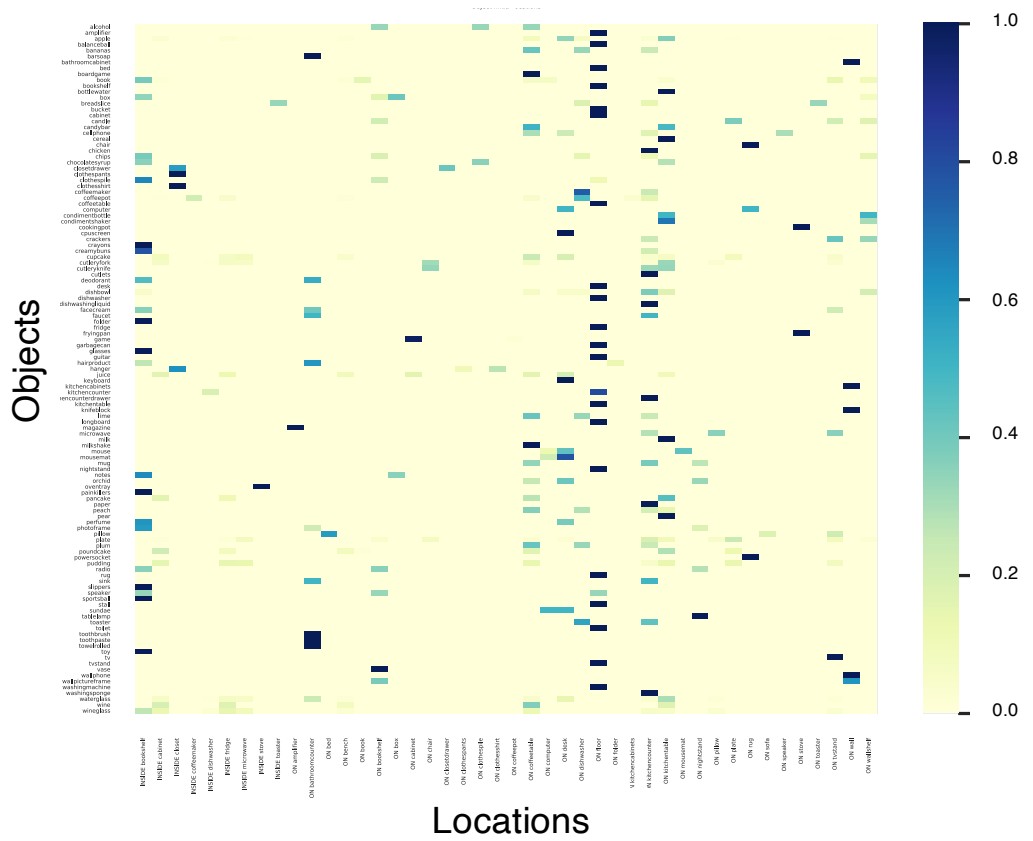

Figure 14: Initial location distributions of all objects in the environment. Rows are objects and columns are locations. The color indicates the frequency.

For training goal inference, we also provide an additional training set of 5303 demonstrations (without pairing helping environments) synthesized in the 5 training apartments. Note that these demonstrations are exclusively used for training goal inference models and would not be used for helping tasks.

### B.3 Distribution of Initial Object Locations

Figure 14 shows the initial location distribution of all objects in the helping environments sampled for the challenge, and Figure 15 shows the initial location distributions for only the objects involved in the goal predicates.

## C Implementation Details of Baselines

### C.1 Goal Inference Module

Figure 16 shows the architecture of the goal inference model described in the paper, where $d = 128$ indicates the dimension of vectors. In this network, the LSTM has 128 hidden units and the MLP units are comprised of two 128-dim fully connected layers. For both node embeddings and the latent states from the LSTM, we use average pooling.

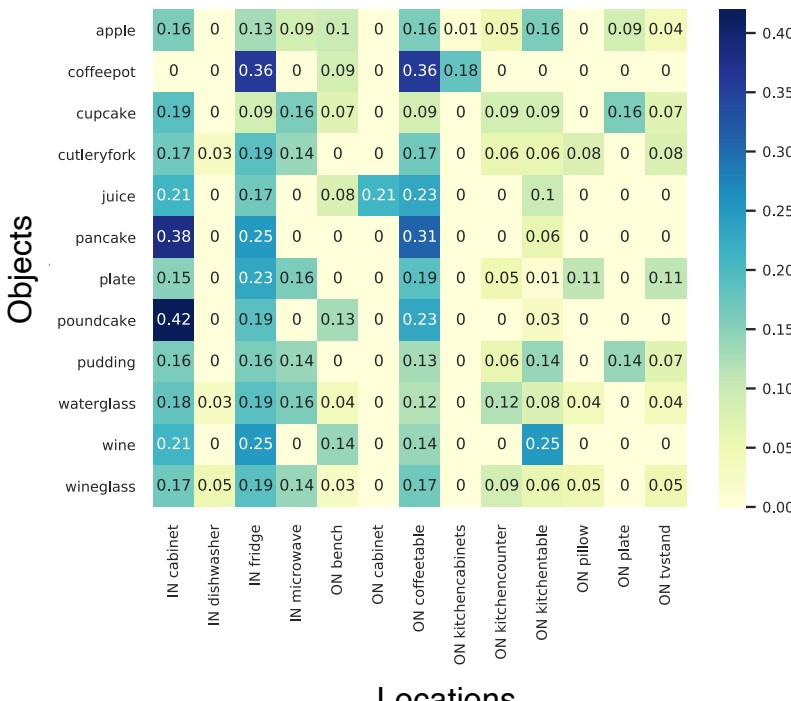

Figure 15: Initial location distributions of the goal objects. Rows are objects and columns are locations. The color indicates the frequency.

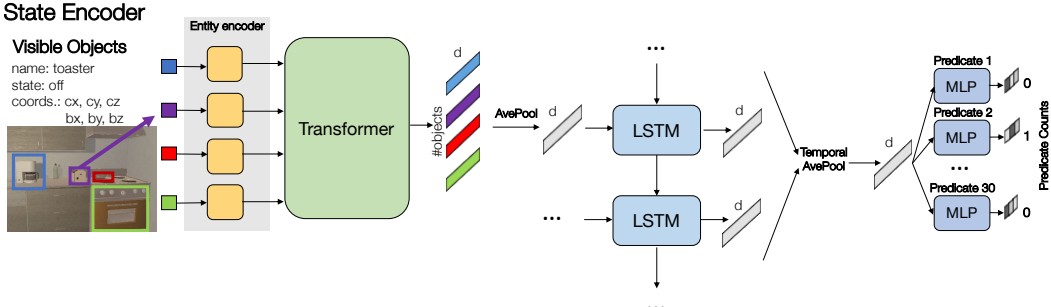

Figure 16: Network architecture of the goal inference model, which encodes the symbolic state sequence in demonstrations and infers the count for each predicate.

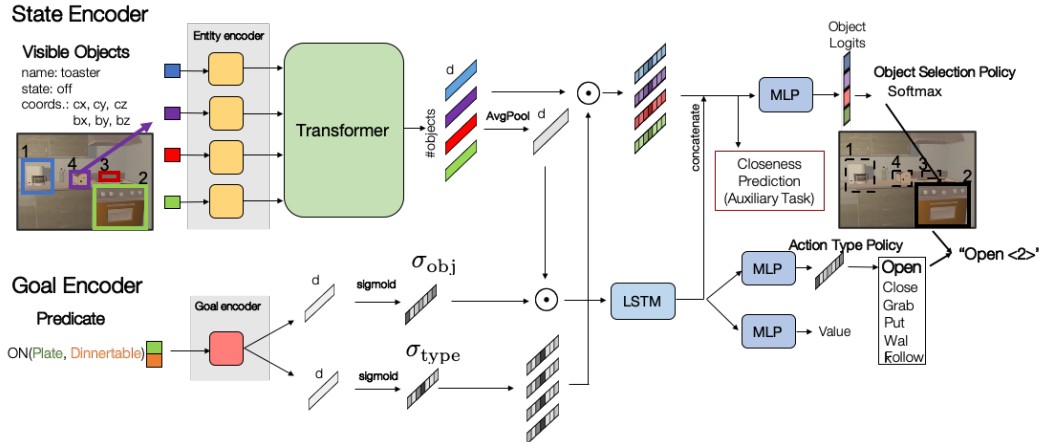

Figure 17: Network architecture of the low-level policy in the **HRL** baseline. Note that the object selection policy also considers "Null" as a dummy object node for actions that do not involve an object, which is not visualized here.

## C.2    HIERARCHICAL PLANNER

The hierarchical planner (**HP**) baseline is similar to the planner designed for the human-like agent (Section A.3) but has its own observation and belief. When given the ground-truth goal of Alice, the MCTS-based high-level planner will remove the subgoal that Alice is going to pursue from its own subgoal space.

## C.3    GENERAL TRAINING PROCEDURE FOR RL-BASED APPROACHES

We train the high-level RL policy by giving ground-truth goals and by using RP as the low-level planner to reach the subgoals sampled from the high-level policy. Whenever a goal predicate is satisfied (either by Alice or by Bob), Bob will get a reward of +2; it will also get a -0.1 penalty after each time step. We adopt the multi-task RL approach introduced in Shu et al. (2017) to train the low-level policy in a single-agent setting, where we randomly sample one of the predicates in the goal in each training episode and set it to be the objective for Bob. This is to ensure that Bob can learn to achieve subgoals through the low-level policy by himself. The **HRL** baseline is implemented by combining the high-level and low-level policies that are trained separately.

## C.4    LOW-LEVEL POLICY

Figure 17 illustrates the network architecture for the low-level policy. We use the symbolic observation (only the visible object nodes) as input, and encode them in the same way as Figure 16 does. We encode two object classes in the given subgoal $sg$ (i.e., a predicate) through word2vec encoding yielding two 128-dim vectors. We then concatenate these two vectors and feed them to a fully connected layer to get a 128-dim goal encoding. Based on the goal encoding, we further get two attention vectors, $\sigma_{\text{object}}$ and $\sigma_{\text{type}}$. Each element of the attention vectors ranges from 0 to 1. For each object node, we use the element-wise product of $\sigma_{\text{object}}$ and its node embedding to get its reshaped representation. Similarly, we can get the reshaped context representation by an element-wise product of the context embedding and $\sigma_{\text{type}}$. This is inspired by a common goal-conditioned policy network architecture (Chaplot et al., 2018; Shu et al., 2017), which helps extract state information relevant to the goal. From each reshaped node representation, we can get a scalar for each object representing the log-likelihood of selecting that object to interact with for the current action. After a softmax over all the object logits, we get the object selection policy $\pi_{\text{object}}(k|o^t, sg)$, where $k$ is the index of the object instance selected from all visible objects (which also includes "Null" for actions that do not involve an object). For encoding the history, we feed the reshaped context representation to an LSTM with 128 hidden units. Based on the latent state from the LSTM, we get i) the action type policy $\pi_{\text{type}}(a|o^t, sg)$, which selects an action type (i.e., "open," "close," "grab," "put," "walk,"

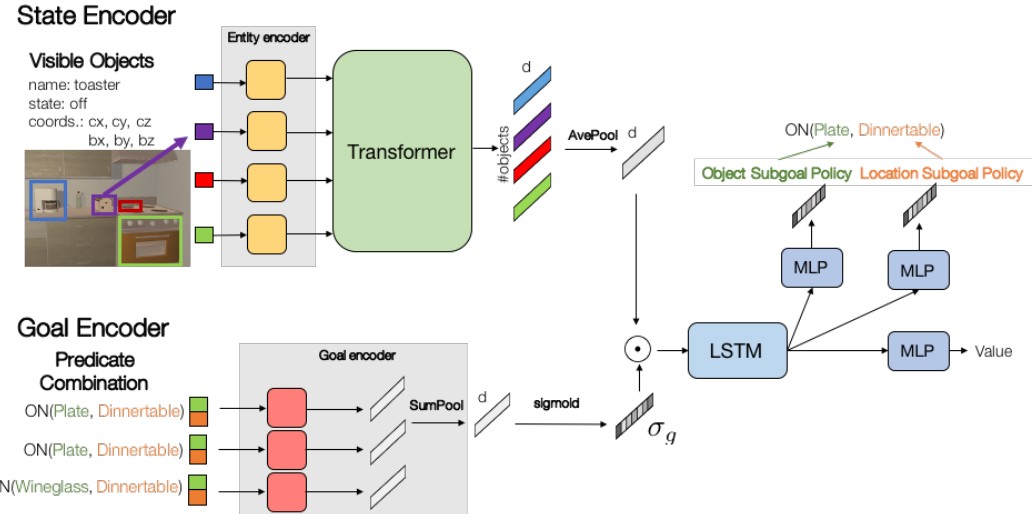

Figure 18: Network architecture the high-level policy for the **Hybrid** and the **HRL** baselines.

or "follow"), and ii) the value function $V(o^t, sg)$. The sampled $k$ and $a$ jointly define the action for the AI agent. Note that some sampled combinations may not be valid actions, which will not be executed by the VirtualHome-Social environment.

In addition to the policy and value output, we also build a binary classifier for each visible node to predict whether it is close enough for the agent to interact with according to the symbolic graphs. This closeness prediction serves an auxiliary prediction which helps the network learn a better state representation and consequently greatly improves the sample efficiency.

In each training episode, we randomly sample a predicate from the complete goal definition as the final goal of the agent. The agent gets a reward of 0.05 for being close to the target object and/or location, and a reward of 10.0 when it grabs the correct object or puts it to the correct location. Note that when training the low-level policy, we set up a single-agent environment to ensure that the AI agent can learn to achieve a predicate by itself.

We adopt a 2-phase curriculum learning similar to Shu et al. (2017): In the first phase, we train a policy for grabbing the target object indicated in the goal. During this phase, a training episode terminates whenever the agent grabs the correct type of object. In the second phase, we train another policy which learns to reuse the learned grabbing policy (which is deployed whenever the "grab" action type is sampled) to get the goal object and then put the grabbed object to target location specified in the goal.

We use off-policy advantage actor-critic (A2C) (Mnih et al., 2016) for policy optimization. The network is updated by RMSprop (Tieleman & Hinto, 2012) with a learning rate of 0.001 and a batch size of 32. The first phase is trained with 100,000 episodes and the second phase is trained with 26,000 episodes.

### C.5 HIGH-LEVEL POLICY

As Figure 18 depicts, the high-level policy (used by **Hybrid** and **HRL** baselines) has a similar architecture design as the low-level policy. Compared with the low-level policy, it does not need to define object selection policy; instead, based on the latent state from the LSTM, it outputs the policy for selecting the first and the second object class in a predicate to form a subgoal[3]. It also augments the goal encoder in the low-level policy with a sum pooling (i.e., Bag of Words) to aggregate the encoding of all predicates in a goal, where predicates are duplicated w.r.t. their counts in the goal definition (e.g., in Figure 18, ON(plate, dinnertable) appears twice, which means there are

---

[3]Note that this is different from the subgoals generated from the high-level planner (Section A.3), which would specify object instances.

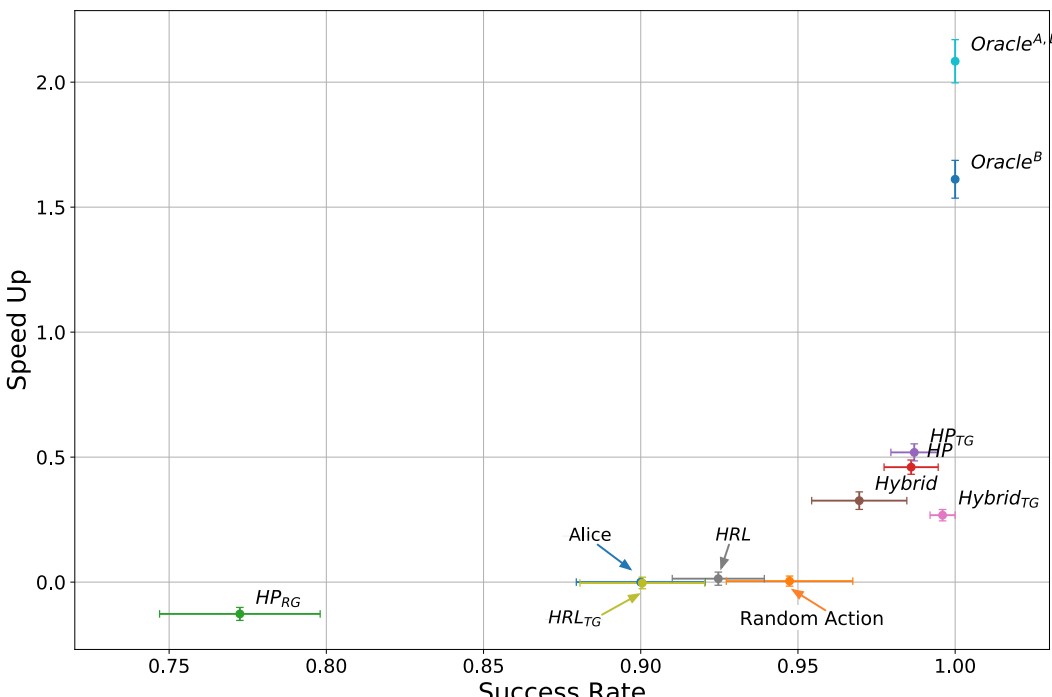

Figure 19: Success rate (x axis) and speedup (yaxis) of all the baselines and oracles

should be 2 plates on the dinnertable). Similar to the low-level policy, we get an attention vector $\sigma_g$ from the goal encoding to reshape the state representation. In total, the network has three outputs: the object subgoal policy for sampling the object class name in the subgoal, the location subgoal policy for sampling the target location class name in the subgoal, and a value function.

The high-level policy is trained with a regression planner deployed to find a low-level plan for reaching that subgoal. Note that the regression planner searches for a plan based on a state sampled from the agent's belief maintained by a belief module discussed in Section A.3. It will also randomly select object instances from the sampled state that fit the defined object classes in the subgoals sampled from the high-level policy.

Similar to the low-level policy, we use off-policy A2C for policy optimization, and the network is updated by RMSprop with a learning rate of 0.001 and a batch size of 16. We first train the high-level policy in a single-agent setting where the AI agent is trained to perform a task by itself; we then finetune the high-level policy in the full training setting where the human-like agent is also present and works alongside with the AI agent. During training, we always provide the ground-truth goal of Alice to the AI agent.

## D    ADDITIONAL DETAILS OF HUMAN EXPERIMENTS

### D.1    HUMAN SUBJECTS

Both the collection of human plans as well as the evaluations in our user studies were conducted by recruited participants, who gave informed consent.

### D.2    PROCEDURE FOR COLLECTING HUMAN PLANS

To collect the tasks for both experiments, we built a web interface on top of VirtualHome-Social, allowing humans to control the characters in the environment. Specifically, the subjects in our human experiments were always asked to control Alice. At every step, humans were given a set of visible objects, and the corresponding actions that they could perform with those objects (in addition to

the low-level actions), matching the observation and action space of the human-like agent. When working with an AI agent, both the human player and the AI agent took actions concurrently.

In both experiments, human players were given a short tutorial and had a chance to get familiar with the controls. They were shown the exact goals to be achieved, and were instructed to finish the task as fast as possible. For each task, we set the same time limit, i.e., 250 steps. A task is terminated when it exceeds the time limit or when all the goals specified have been reached.

The 30 tasks used in the human experiments were randomly sampled from the test set and were evenly distributed across 5 task categories (i.e., 6 tasks for each category).

In Experiment 2, each subject was asked to perform 7 or 8 trials. We made sure that each subject got to play with all three baseline AI agents in at least 2 trials.

### D.3 EXAMPLE OF HUMAN ADAPTING TO AI AGENTS WITH CONFLICTING GOALS

The main reason why real humans work better than the human-like agent when paired with an AI agent that has a conflicting goal (in particular, the $\mathbf{HP}_{\text{RG}}$ baseline), is that they can recognize the conflicting goal, and avoid competing over the same objects forever. Figure 20 depicts an example of this adaptive behavior from a real human player in Experiment 2, which results in the completion of the task within the time limit. Note that in our experiments, a task is considered successful and terminated once all the predicates in a goal have been achieved.

This also calls for an AI agent with the ability to adjust its goal inference dynamically by observing Alice's behavior in the new environment (e.g., Alice correcting a mistake made by Bob signals incorrect goal inference).

### D.4 SUBJECTIVE EVALUATION OF SINGLE AGENT PLANS

To evaluate whether people think the human-like agent behaves similarly to humans given the same goals, we recruited another 8 subjects. We showed each subject 15 videos, each of which is a video replay of a human or the human-like agent performing one of the 30 tasks (we randomly selected one human video and one built-in agent video for each task). For each video, subjects were given the goal and asked to rate how much they agreed with the statement, "the character in the video behaves similarly to a human given the same goal in this apartment," on a Likert scale of 5 (1 is "strongly disagree," 3 is "neutral," and 5 is "strongly agree")[4]. The average ratings for the characters controlled by the human-like agent and by the real humans are 3.38 ($\pm 0.93$) and 3.72 ($\pm 0.92$) respectively. We found no significant difference between the ratings for the human-like agent's plans and the ratings for the real humans' plans in our tasks, as reported by a paired, two-tailed t-test ($t(29) = -1.35$, $p = .19$). This demonstrates that the proposed human-like agent can produce plans that are similar to real humans' plans in our challenge.

Based on the free responses collected from the subjects who rated these videos, human plans look slightly more efficient sometimes since they do not look for objects in unlikely places and avoid moving back and forth between rooms frequently. The human-like agent behaves similarly in most of the time but would occasionally search through the rooms in a counter-intuitive order due to its bounded rationality and the fact that plans are sampled stochastically.

### D.5 ADDITIONAL QUANTITATIVE ANALYSES OF HUMAN EXPERIMENT RESULTS

To evaluate whether the performance of a baseline AI agent helping the human-like agent reflects the performance of it helping real humans, we conduct paired, two-tailed t-test for the three baselines in Experiment 2 based on their cumulative rewards. For $\mathbf{HP}_{\text{RG}}$, there is a significant difference between helping the human-like agent and helping real humans ($t(29) = -2.36$, $p = .03$) as discussed in Section 6 and Appendix D.3. However, there is no significant difference for $\mathbf{HP}$ ($t(29) = -1.78$, $p = .1$) and $\mathbf{Hybrid}$ (($t(29) = -0.5$, $p = .62$)). This validates that, in general, collaboration with

---

[4]Since we focus on the agents' plans in this work, users were instructed to focus on the actions taken by the agents, rather than the graphical display of their body motion.

**Ground-truth goal:**
```
ON(plate, dinnertable): 1
ON(waterglass, dinnertable): 2
ON(wineglass, dinnertable): 1
ON(fork, dinnertable): 2
```

**A random goal sampled by Bob (HP_RG):**
```
IN(wineglass, dishwasher): 1
ON(poundcake, dinnertable): 2
IN(pancake, fridge): 2
ON(wine, dinnertable): 1
```

**The human-like agent and HP_RG**

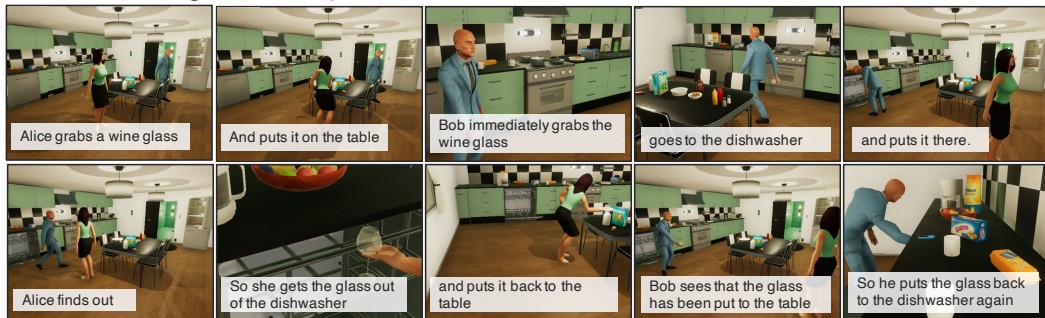

**A real human player and HP_RG**

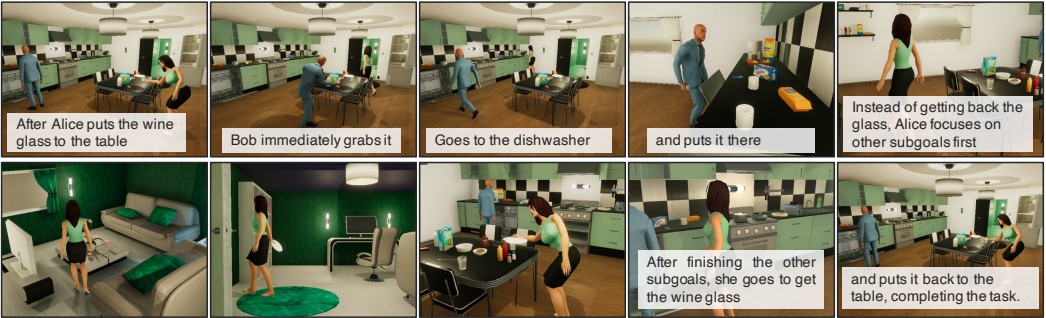

Figure 20: An example of how real human differs from the human-like agent when working with an AI agent (i.e., **HP**_RG) with a conflicting goal. In this example, Bob incorrectly thinks that Alice wants to put the wine glass to the dishwasher whereas Alice actually wants to put it to the dinner table. When controlled by a human-like agent, Alice enters into a loop with Bob trying to change the location of the same object. The real human player, on the other hand, avoids this conflict by first focusing on other objects in the goal, and going back to the conflicting object after all the other goal objects have been placed on the dinner table. Consequently, the real human completes the full task successfully within the time limit.

the human-like agent is comparable to collaboration with real humans. Given these analyses, the training and evaluation procedure[5] presented in this paper is both scalable and comprehensive.

---

[5]I.e., i) training AI agents with the human-like agent, and then ii) evaluating them both with the human-like agent (in a larger test set), and with real humans (in a smaller but representative test set).

