# OpenReview forum: "Watch-And-Help: A Challenge for Social Perception and Human-AI Collaboration"
_ICLR.cc/2021/Conference — ICLR 2021 Spotlight_

### Official Review · AnonReviewer3 · 2020-10-21
**Interesting paper with a lot of good ideas, somewhat let down by its ambitious scope and large amount of content. A smaller scope and tighter focus would improve this paper significantly.**

**Rating:** 6
**Confidence:** 4

**Review:**

### Quality
- I enjoyed reading the paper. In general I think it was of good quality, but mentioned in my summary there was a lot going on from lots of perspectives. T
- This not only includes the scope but the experiments (agent and humans), large number of complex agent technologies, many types of agents.
- Obviously the page count would not suffice and hence there is what I would consider significant core content in the appendices.
- While the conclusion is short and to the point, the paper would have benefited from a critical evaluation/discussion of the approach taken (limitations etc).
- In the conclusion the authors state that the experiment demonstrated the ability to evaluate aspects of social intelligence at scale. What do you mean by scale? What if I had 10, 100, 1000 agents?

### Clarity
- From a readability point of view the paper is largely OK.
- The abstract and introduction which motivate the problem are very good.
- I would have liked to have seen further
- Lots of moving parts/components/subsystems made it sometimes difficult to get your head around the whole system being described.
- It wasn't clear to me as to the significance of the domain chosen. What is the long term goal with this work? Is it social intelligence for human robotics?
- I think the authors could do a better job of explaining/interpreting results.

### Originality
- There were certainly many original aspects to the work. But I believe it was more combining existing approaches and techniques to solve the problem rather than coming up with new learning techniques from scratch.

### Significance
- The work is significant to researchers who are interested in modelling social intelligence, robotics and multi-agent systems.
- So perhaps with that in mind this paper may be more suited to a conference like AAMAS (Autonomous Agents and Multi-Agent Systems) rather than ICLR which is more machine learning focused. I'm not making a judgement here, just something to consider with this work going forward.

### Other questions and comments for the authors:

- Does Bob learn the sequence of actions that Alice take to achieve the task or just the end state of Alice's task? If it is the latter, does Bob work as well if he doesn't participate in the watch stage and is just told what the goal is in the help stage?

- If Bob is helping Alice, does he automatically assume that Alice is trying to achieve the task that Bob watched Alice previously doing? Or does Bob have to observe Alice a second time and then infer that she is trying to achieve a goal which Bob knows how to help with because he previously observed her doing it in the Watch phase?

- In the watch phase, does Bob ever get confused? For example, assuming that Alice has started task A, when in fact she started task B and Bob starts helping with task A?

- You don't mention agent communication which is often an important part of modelling agent social interaction. What if Alice communicated to Bob her goal?

- I really like Figure 6a.This is an excellent chart except for the y-axis removal of sections between 0.6 and 1.6 and 1.8 and 2.0. I think this is misleading. Perhaps a log scale would be appropriate.

- That aside I think this type of chart tells the reader more about the performance of the different agents compared to 6b which plots the reward. With 6b I need some additional interpretation which is a bit lacking in the text.

- What are the error bars in 6b? Standard deviation, standard error, confidence intervals?

- While the addition of human experiments is good and comprehensive, perhaps that could be the topic of another paper and in its place have section on discussing the pros/cons of the results.

---

> ### Author Response · Authors · 2020-11-19
> **Response to Reviewer #3**
>
> Thank you for your comments and helpful suggestions, we are encouraged that you value the clarity and motivation of the challenge. We address your specific questions and concerns below.
>
>
> **Evaluation at scale**
>
> While the challenge allows to have multiple agents playing the role of Bob, here we focus on using a single agent. We talk about evaluation at scale referring to the ability to evaluate the challenge within a large set of environment conditions and goals, given the ability to generate new scenarios with the simulator, and the objective evaluation metrics.
>
> **Interpretation of the results**
>
> We have extended the discussion of results in Section 5.3 of the updated version.
>
> **Originality**
>
> Indeed, the goal of this work is to present an environment and benchmark, with evaluation metrics to measure progress in the defined collaboration task. We provide non-trivial baselines that use existing approaches to predict goals during the *Watch* phase as well as finding a planning policy on the *Help* phase, showing that these leave a margin of improvement in future methods.
>
> **What does Bob learn about the task during the Watch phase**
>
> During the *Watch* phase, Bob has access to a sequence of partial observations from Alice’s demonstration. Learning from the end state is not enough, since it may omit non visible objects that are relevant to the task. We report the performance when Bob is given the true goal for all our baselines, indicated by the subscript $_{TG}$. Using the true goal helps improve the success rate of the task, as well as the speedup.
>
> **Bob’s knowledge and behavior during the Help phase**
>
> In the challenge, each demonstration from the *Watch* phase is paired with an environment from the *Help* phase. The Helper model is designed assuming that the goal of the task is the same as the one observed during the Watch phase, though in an unseen environment. Bob does “get confused” in certain episodes, when the inferred goal is different than the one Alice was performing. This effect is much more evident in the baseline using a Bob that selects random goals during the Watch phase ($HP_{RG}$). Despite relying on the symbolic planner, the most powerful baseline, helping in the wrong goals is detrimental to the collaboration, with negative speedups and lower success rate compared to the single agent Alice. An example of that instance is shown in Fig. 19.
>
> **Agent communication**
>
> In this work, we focus on non-verbal communication, which is commonly used in human cooperation. Warneken et al. [1] show that infants collaborate with no explicit communication or directives from adults. In [2] there is a thorough discussion on the importance of non-verbal communication in humans.
> We want to note that VirtualHome-Social can include communication with minimal extensions, though doing it would change the conditions of the challenge.
> If Alice communicated to Bob her goal, we would see the same effect as our True Goal baselines, with the *Watch* stage being unnecessary to solve the task. We could also put a cost on communication, forcing agents to communicate efficiently in order to perform the goal. Agents could then communicate the general goal, without referring to the exact predicates that need to be achieved, or send information about their beliefs and intentions. While this opens the possibility of different paradigms of collaboration, we believe it is out of the scope of the current work, and rather a direction of future work, as stated in the conclusion.
>
> **Plots in Figure 6**
>
> Thank you for your suggestions. We decided to break the axis to show more clearly the performance of the non-oracle baselines. We have included in the supplementary materials a version without breaking the axes, showing more clearly the gap to the oracle baselines. We included plot 6.b to provide a single measure that could combine success and the time it takes to complete different tasks for the different baselines. This allows us to compare the performance of different kinds of goals in a more compressed way. The error bars in both plots correspond to the standard error.
>
> **Human experiments**
>
> Thank you for your suggestions. While further experiments with humans would be a topic for a new paper, we believe it is important to include the human experiments we present in the paper. The goal of our work is to study aspects of Human-AI collaboration in a scalable scenario, and these experiments provide a way to validate our results, measuring (i) how similar is the human-like agent to real humans and (ii) whether humans benefit similarly from the designed helper agents. These experiments also allow us to extend our evaluation to measures that are difficult to numerically quantify, such as trustiness in the agents or helpfulness.
>
>
> [1] Felix Warneken and Michael Tomasello. Altruistic helping in human infants and young chimpanzees. Science, 311(5765):1301–1303, 2006.
>
> [2] Tomasello, M.. “Origins of human communication.” (2008).

---

### Official Review · AnonReviewer2 · 2020-10-26
**Clear, novel, empirically rigorous, and highly reproducible work. Recommend accept.**

**Rating:** 7
**Confidence:** 3

**Review:**

	1. Summary of Paper:
		1. This paper contributes a new challenge task (called Watch and Help) and a multi-agent virtual environment evaluation platform (a multi-agent extension of VirtualHome called VirtualHome-Social) for goal recognition and collaborative planning. The Watch and Help task requires an evaluated agent (say Bob) to watch a single demonstration of another agent (say Alice) performing a task in isolation. These tasks are drawn from a set of 5 potential home-related task sets (e.g. reading, putting groceries away, etc.). After watching Alice perform a given task in isolation, Bob is required to help Alice perform that same task in a different environment. Bob's help or the success of the collaboration is measured by the success rate, speed up (vs. performing the task alone), and cumulative reward (+1 if goal is successful and -1 if goal is not after T timesteps) from performing the task. The article presents a "human-like" planning agent that uses hierarchical planning (MCTS search for sub-goal selection and regression planning for low-level action selection) over beliefs and goal states to act as the fixed demonstrator and collaborator to train the agents being evaluated. The paper also provides several benchmark agents and compares their collaborative performance with both the hierarchical planning Alice agent and a human-controlled Alice avatar.
	2. Strengths
		1. Clarity:
			a. The paper is remarkably clear while explaining its contributions, the challenge task, agent behaviours during experiments, experimental comparisons, etc.
			b. The paper uses visualisation figures and images from the environment to great effect explaining sequences of behaviour that would otherwise be difficult to show without the reader having to look at supplementary videos (which are nonetheless provided).
			c. The related work is clearly specified showing how relevant works differ from the current approach. One potential weakness in comparisons to comparable virtual environments is given later.
			d. The different variant agents being compared in the task benchmark are clearly explained and form an appropriate experimental design.
		2. Novelty/Impact
			a. The work aims to showcase a new challenge task, evaluation platform, and benchmark agent performance for goal recognition followed by collaborative planning. The work as described compares favourably to similar work in evaluation platforms and benchmarks referenced in the related work section and appendix. The differences are made clear, though the use of some featured distinctions are not demonstrated in the paper (e.g. visual observations are possible but not used in the benchmarking).
		3. Experimental Rigour
			a. This work is not primarily about demonstrating the benefits of a particular approach over others in a particular application. It demonstrates benchmarks for agent performance in a newly introduce problem setting. From that perspective, the paper has strong experimental rigour. The experimental design is appropriate, comparing multiple baselines and oracles with several sets of experimental variants from both automated planning and reinforcement learning communities.
			b. The comparison of experimental variants is conducted with both a computationally-controlled agent and a human-controlled avatar to evaluate collaboration performance in increasingly realistic settings.
			c. The claim that the computationally-controlled Alice agent is human-like is repeated throughout the paper. This is not justified in the actual text of the main paper, but is supported to a moderate degree (human-like in strategy/planning if not movement/behaviour) through experiments with human subjects that are described in the appendix.
			d. Effort paid to ensure diverse avatars in experimentation.
		4. Reproducibility
			a. The work is almost entirely reproducible, with details of all agent architectures used for experiments provided with hyperparameters and architecture design. The authors describe that the environment will be released as open-source, which will then make the article wholly reproducible. This reviewer appreciated the level of additional detail provided in the appendix to improve this area of evaluation.
	3. Weaknesses
		1. This paper uses the term social intelligence to motivate the context for this challenge task. Social intelligence is a much broader term than what is actually being evaluated here and would require evaluating capabilities beyond goal recognition and coordinated planning/task execution. It is suggested to replace this claim with "goal recognition and collaborative planning".
		2. From the motivation provided, i.e. evaluating social perception and collaboration, why is the task specifically about watching then helping and not both together or just helping or just watching or some combination of these activities fluidly occurring throughout the interaction?
		3. Further, the work itself does not explicitly motivate why this, specific challenge task for goal recognition followed by collaborative planning is necessary for moving the state of the art in human-AI collaboration forward. However, it is a small leap to see the impact of this platform/task in evaluating applications like service robotics, social robotics, collaborative human-agent task performance, video games, etc. This reviewer can understand the impact of the work, but it would be clearer to explicitly discuss this.
		4. It would be clearer to specify that this task is limited to situations where there is explicitly only one goal throughout the entire demonstration + execution episode. This is important since it precludes using this challenge task for research into agents that need to use goal recognition after the initial demonstration, potentially continuously over the course of execution. This second kind of continuous goal monitoring is more similar to real-world applications of watching and helping or assistive agents or social robotics, since the human collaborator can (and often will) change their mind.
		5. Similarly, it should be noted that there is an explicit limitation of this challenge task and the evaluation metrics to scenarios where the entire success or failure of the approach is purely based on the final team accomplishment. This is similar to situations like team sports, where all that matters is the final game score. Many real-world scenarios for human-AI collaboration, differ by also requiring individual collaborators to do well or for the primary human user to do better with collaboration (than without). For example, in a computer game where Bob represents a team-mate to Alice who is a human player, Bob can choose to steam-roll Alice and win the game by itself. However, this leads to lower subjective user experience for the human team-mate. In this case, the score might be greater than what Alice could accomplish on their own and the game might be won faster than Alice could on their own, but the experience would be different based on whether they are truly collaborating or one is over-shadowing the other.
		6. A final assumption, is that there is no difference in expertise between Alice and Bob. The human is expected to be able to competently finish the task and for Bob to linearly cut down the time taken to perform this task. There are many real-world tasks in human-AI collaboration where this assumption does not hold and there could be non-linear interactions between success-rate and speed-up due to different levels of expertise between Alice and Bob.
		7. The fixed Alice agent is called human-like through out the article and this was not properly justified anywhere in the main text of the paper. However, the appendix actually describes results that compare the performance of the computationally-controlled and human-controlled variants of Alice to human observers. This potentially justifies this weakness. For clarity, it would be valuable to refer to the presence of this validation experiment in the main paper.
		8. Why aren't there benchmark results (more than one) for the goal recognition component similar to the planning task experimentation? If both parts of the task are important, it would be valuable to provide additional experiments to show comparisons between goal recognition approaches as well, even if that is in the appendix for space reasons.
		9. There could be more analysis of the benchmark agent performance, 1) Why does the purely random agent work relatively well across tasks? 2) Why doesn't HRL work better? Is this due to less hyperparameter tuning compared to other approaches or due to some intrinsic aspect of the task itself? 3) Perhaps I missed this, but why not try a flat model-based or model-free RL without a hierarchy?
		10. There are several comments about other environments in the related work section and appendix being toy environments. However, the tasks in the environment demonstrated in this paper only use a small set of predicates as goals. Similarly, it CAN generate visual observations but that isn't used by any of the baselines in the paper. Several comparisons to related virtual environments are made in appendix, but some of the features aren't used here either (humanoid agent - this challenge task works equally well with non-humanoid avatars/behaviours and realism - visual realism is present but it isn't clear if behavioural or physical realism is present due to seeming use of animations instead of physical simulation).
		11. None of the tasks described allow the use of communication between agents or evaluate that. Other multi-agent environments like Particle Environments (below) allow for that. Communication is a natural part of collaboration and should have been mentioned if only to distinguish future work or work out of current scope.
			a. @article{mordatch2017emergence,  title={Emergence of Grounded Compositional Language in Multi-Agent Populations},  author={Mordatch, Igor and Abbeel, Pieter},  journal={arXiv preprint arXiv:1703.04908},  year={2017}}
		12. "planning and learning based baselines", "and multiple planning and deep reinforcement learning (DRL) baselines", etc. - There is potential for confusion with the use of terms "planning" and "learning" methods to do what both fields (automated/symbolic planning and reinforcement learning) would potentially consider as planning tasks. It would be clearer to indicate this distinction in terminology.
		13. The human-likeness evaluation experiment asked subjects to evaluate performance one agent video at a time. A more rigorous evaluation might compare two agents side by side and ask the human to guess the human performance. This could also be in addition to the current evaluation. The current evaluation is a ceiling on performance while the comparative evaluation is a potential floor.
	4. Recommendation:
		1. I recommend accepting this paper, which was clear, novel, empirically strong, and supremely reproducible. The strengths conveyed above outweighed the weaknesses.
	5. Minor Comments/Suggestions:
		1. Some minor typos in the manuscript:
			a. Using the inferred goals, both HP and Hybrid can offer effective. - page 6
			b. IN(pundcake, fridge) - appendix table 2
			c. This closeness perdition - appendix page 19

---

> ### Author Response · Authors · 2020-11-20
> **Response to Reviewer #2 - Part 1.**
>
> Thank you for your comments and helpful suggestions, we are encouraged that you value the novelty, clarity and reproducibility of the challenge. We address your specific questions and concerns in the 2 comments below.
>
> **1. On open-sourcing**
>
> We will open source the code, data and models for the challenge and environment. We include in the general response the source code of the benchmark, as well as an executable of the environment to run the baselines described in the paper.
>
> **2. Using Social Intelligence**
>
> We agree that Social Intelligence includes many aspects and important problems. As we state in the paper, we address some of them in our challenge. We think it is important to still frame the problem in the domain of Social Intelligence, given that successful agents need to infer the beliefs and intentions of the other agents they are collaborating with. Moreover, the human quality measures reward not only fast collaboration, but also other factors such as humans’ trust of AI agents and their perception of AI agents’ helpfulness.
>
> **3. Motivation for the challenge and justification of separate Watch and Help stages.**
>
> The proposed two-stage setup has been studied in developmental psychology [1], where infants are shown a demonstration of the task done successfully before evaluating their abilities to help. Our challenge is designed to match this setup, separating the inference of the goal and the collaboration. Moreover, this setup allows us to test how Bob can help in a new environment that has not been seen during the Watch stage.
> Nevertheless, performing goal inference and collaboration in a single stage is a great suggestion, and would require a new methodology, so that agents could balance between gathering information about the goal and collaborating on the task. We leave this as a direction for future work.
>
>
> **4. Challenge limited to situations where there is a single goal**
>
> Since each task is defined by a set of predicates, it is possible to define tasks with multiple goals where predicates are sampled from multiple predicate sets corresponding to different activities.
>
> For this, we create another test set, where each task has goals for two activities (e.g., putting groceries to the fridge and setting up a dinner table), and evaluate the goal inference model that is trained on single goals. We also evaluate the performance of Alice alone as well as Alice with the best performing Bob (HP). We ensure that the combined goals do not have conflicting predicates, such as setting a table and putting dishes in the dishwasher. The goal inference model reports a precision and recall on the multi-goal tasks of 0.68 and 0.64 respectively. Alice alone achieves a success rate of 95.40 (SE=.01). When the symbolic planner Bob collaborates, the task has a speedup of 0.21 (SE=0.04) and a success rate of 88.60 (SE=.02). Note that given the lower performance of the goal inference model, it is more likely that Bob ends up doing conflicting actions, as shown in the paper for the random goal baseline.
> We have included these additional results in the paper.
>
> **5. More diverse evaluation metrics**
>
> As noted by the reviewer, there are many important factors in evaluating Human-AI collaboration, many of which are difficult to quantify. The proposed challenge focuses on a metric that is quantifiable and thus allows us to measure the progress in a systematic and reproducible way. We argue that the proposed metrics are informative, since in order to achieve high scores (reward, success rate, and speedup) in the proposed tasks, agents need to distribute labor intelligently and reason about the others’ goals and beliefs.
>
> We want to underline the importance of other types of metrics despite the difficulties in their implementations. This is what motivates our human studies, where we include qualitative measures of Trust, Helpfulness and Perceived Understanding of the Goal, similar to [2].
>
> **6. Difference of expertise between the agents**
>
> Incorporating different levels of expertise is a great suggestion, as it would require Bob to not only understand the goal but also infer the expertise of Alice to help effectively. We leave this direction for future work. We want to note though, that in our evaluation we provide a measure of expertise via the Oracles, that measure performance when Bob alone, or Bob and Alice have full observability of the environment.
>
>
> **7. Supporting the claim of a Human-like agent**
>
> Thank you for the suggestion. We will include in the revised version a note to the validation experiment, supporting the claim of the human-like agent terminology.

---

> > ### Author Response · Authors · 2020-11-20
> > **Response to Reviewer #2 - Part 2.**
> >
> > We address here further questions concerns, and provide the references mentioned in the current and previous response.
> >
> >
> > **8. Benchmark results on Goal Inference**
> >
> > We have included in the revised paper two variants for the goal recognition model. In particular, we experiment with using the last state of the LSTM to predict the predicates, as well as the ground-truth actions from Alice. The model using the last state obtains a precision and recall of 0.70 and 0.57, whereas the model with ground-truth actions obtains a precision and recall of 0.99 and 0.99 respectively. As a reference, the precision and recall of a random goal baselines is 0.08 and 0.09. We will include more extensive baselines on the final version of the paper.
> >
> > **9. Analysis of the benchmark**
> >
> > We include a more thorough analysis in the revised version of the paper. We also answer the specific questions brought up.
> >
> > (a) **Performance of a purely random agent:** We want to note that the random agent is still performing the tasks with Alice, which is not random and has knowledge of the goal of the task. For this reason, there should be no difference between using a Random agent to help or not using any agent at all. In the results section we confirm that, by showing no significant difference between the random agent and Alice alone, as revealed by a two-tail t-test (t(99) =−1.38,p= 0.17).
> >
> > (b) **Model-Free RL and HRL:** We experimented with a flat model-free RL policy as a baseline, but it failed to learn any successful behavior, despite experimenting with different partial rewards and curriculums. We note that, even in the hierarchical case, the low level policy was trained in a 2-phase curriculum learning scheme, as described in C.4, and the results of the HRL policy were not better than Alice alone.  This is due to the large horizon of the defined tasks, and the complexity of the action and observation space in the environment, where the actions change according to the current observations.
> >
> > **10. Visual Realism**
> >
> > a. In the generated data, we include actual video frames for goal inference. Our API also allows agents to receive raw pixels as inputs instead of the ground-truth states. In the current work, we did not investigate visual perception problems for both watching and helping and focus on modules that come after basic visual perception instead, but it does not prevent one from trying to solve the challenge by integrating a visual perception module into the AI agents.
> >
> >
> > b. Even when not using raw pixels as inputs, we still need physical simulations in order to update ground-truth object and agent states. This includes (i) updating the parietal observations with limited field of view and occlusion, and (ii) executing actions with physical constraints (e.g., we need to use inverse kinematics to physically solve grabbing actions and will report failure if the agent can not reach the target object at the current position; we will also put down the object to an actual physical position when placing actions are called and executed, and will report failures if for instance the target position is too crowded).
> >
> >
> > **11. Communication**
> >
> > Communication is indeed a very important aspect of Human-AI collaboration, and the current platform requires very minimal changes to include it in the environment. While this allows for interesting research directions - including grounding language to the environment or the agents’ beliefs, emergence of language - it would require redefining our measures of performance and largely broaden the scope of this work. Here, we focus on inferring the goals and plans of other agents with non-verbal communication, which is widely used in cooperation between humans [3,4]. We leave direct communication as a future work direction, as stated in the conclusion.
> >
> >
> > **12. On the use of planning and learning-based terms**
> >
> > Thank you for the suggestion, we will clarify the terminology in the paper.
> >
> > **13. Comparative evaluation in human-likeness performance**
> > This is a great suggestion. We will include that as a measure to evaluate how realistic the proposed agent is.
> >
> >
> > We also corrected the typos raised in the manuscript.
> >
> > [1] Felix Warneken and Michael Tomasello. Altruistic helping in human infants and young chimpanzees. Science, 311(5765):1301–1303, 2006.
> >
> > [2] Liu, Chang, et al. "Goal inference improves objective and perceived performance in human-robot collaboration." arXiv preprint arXiv:1802.01780 (2018).
> >
> > [3] Felix Warneken and Michael Tomasello. Altruistic helping in human infants and young chimpanzees. Science, 311(5765):1301–1303, 2006.
> >
> > [4] Tomasello, M.. “Origins of human communication.” (2008).

---

### Official Review · AnonReviewer4 · 2020-10-28
**Interesting AI challenge, but enough contribution for a regular paper for ICLR?**

**Rating:** 6
**Confidence:** 3

**Review:**

This paper introduces an AI challenge where an AI agent needs to collaborate with a human-like agent to enable it to achieve the goal quicker. Two stages are defined: a watch stage and a help stage. In the first one, an AI agent  watches a human-like
agent performing a task once and infers human-like agent’s goal from its actions. In the help stage, the AI agent helps the human-like agent to achieve the same goal in different environments as fast as possible.


This work tackles problems related to human-AI collaboration, a topic that poses several great challenges for the AI community, and other scientific communities too. The main theme of the paper and research presented fits well ICLR.

From my point of view, the authors do an excellent job presenting the challenge, the multi-agent virtual platform (open source planned for the future if I understood correctly?) and all their functionalities (create agents that emulate human behaviours, the interface that supports evaluation, etc.) They also present a benchmark.

However, I am not familiar with having a paper that has as main contribution a challenge. I think there are other contributions in the paper, but they are all presented in terms of such a challenge and the virtual environment. I found myself unsure if the authors evaluate a particular method (that the scientific community can use) or an instantiation of a method that is only valid in the virtual environment/challenge they present. To summarise, either the focus is on the main new ideas presented in the paper and the challenge/virtual environment to demonstrate those, or we make this virtual environment open source (as the plan is) and present the paper not as a regular paper, but another type of paper that describes the virtual environment (including the previous version) as a testing platform for other researchers.

---

> ### Author Response · Authors · 2020-11-19
> **Response to Reviewer #4**
>
> Thank you for your comments and helpful suggestions. We address your specific questions and concerns below.
>
> **1. Paper contributions and fit to ICLR**
>
> **Contributions**: We believe that providing a platform and benchmark for evaluating Human-AI collaboration is a relevant contribution for this conference. We have evaluated common approaches (symbolic planning and RL) as non-trivial baselines for the challenge. By doing so, we show that there is a large performance gap between the existing methodologies and the oracles. Based on the empirical results, we have further discussed the specific problems for human-AI collaboration which our challenge poses for the future work -- for instance, false beliefs, multi-level planning, generalization to unseen environments, recursive reasoning for predicting teammates’ actions.
>
> These problems, as we stated in the paper, are commonly shared by different types of human-AI collaboration (prior work such as Albrecht & Stone, 2018, cited in the paper, has also raised these problems). Here, we are showing a systematic and scalable evaluation on AI agents’ abilities to solve these problems. By building AI agents that can achieve success in the proposed challenge, we can help make progress on building socially intelligent AI agents for human-AI collaboration in general.
>
> **Previous benchmarks in ICLR**: We would like to note that there have been papers accepted to ICLR ( e.g., [1,2,3,4]) focusing on new platforms and challenges as their main contributions. Like ours, the value of this type of work is (i) to raise new problems that existing challenges neglect but are critical for building intelligent systems in general, and (ii) to provide systematic ways to monitor progress on solving these problems. Following your advice, we have updated the paper to include a more thorough discussion on the results and moved part of the description of the platform into the appendix.
>
> **2. Open sourcing the platform**
>
> We will open source the environment as well as the code and models for the challenge, so that other researchers can build models for the proposed benchmark, or extend the platform with new aspects of Human-AI collaboration. We include in the general response the source code of the benchmark, as well as an executable of the environment to run the baselines described in the paper.
>
>
> **References**
>
> [1]  Kexin Yi*, Chuang Gan*, Yunzhu Li, Pushmeet Kohli, Jiajun Wu, Antonio Torralba, & Joshua B. Tenenbaum (2020). CLEVRER: Collision Events for Video Representation and Reasoning. In International Conference on Learning Representations.
>
> [2]  Rohit Girdhar, & Deva Ramanan (2020). CATER: A diagnostic dataset for Compositional Actions & TEmporal Reasoning. In International Conference on Learning Representations.
>
> [3]  Eleni Triantafillou, Tyler Zhu, Vincent Dumoulin, Pascal Lamblin, Utku Evci, Kelvin Xu, Ross Goroshin, Carles Gelada, Kevin Swersky, Pierre-Antoine Manzagol, & Hugo Larochelle (2020). Meta-Dataset: A Dataset of Datasets for Learning to Learn from Few Examples. In International Conference on Learning Representations.
>
> [4]  Weihao Yu, Zihang Jiang, Yanfei Dong, and Jiashi Feng 2020. ReClor: A Reading Comprehension Dataset Requiring Logical Reasoning. In International Conference on Learning Representations.

---

> > ### Comment · AnonReviewer4 · 2020-11-20
> > **Changed my rating to 6: Marginally above acceptance threshold**
> >
> > Dear authors,
> >
> > Thanks for your clarifications and for considering my comments; I do understand now better the aim and presentation of your work. I have changed my rating to above acceptance.
> >
> > Best,

---

### Official Review · AnonReviewer1 · 2020-10-29
**Interesting set up and prototype; should be revised to fit the scope of a conference paper presentation**

**Rating:** 6
**Confidence:** 5

**Review:**

The paper targets to demonstrate social perception and human-AI collaboration in common household activities. It shows the development of a multi-agent virtual environment that is used to test an AI agent’s ability to reason about other agents’ mental states and help them in unfamiliar scenarios. This is performed by presenting an experimental study over specifically selected scenarios which involve aspects of social intelligence.
The set-up includes a watch step, based on goal inference modelling and of a help stage, based on multiple planning and reinforcement learning. This is an interesting set up which can model some aspects of human behavior. The implementation description, which involves specific activities and their analysis and planning, illustrates the presented concepts and methodology.
However, the presentation, in the form of large descriptions, in the main part of the paper, the Appendices and the Supplementary material do not match the requirements of a conference paper; it is rather the presentation of a platform, that would fit a demo of a prototype or a submission to a journal. There is no discussion, justification, or evaluation of the specific algorithmic choices in modules, e.g., in the goal inference module. In fact the paper should be reduced in length, focusing on the AI related issues and selections adopted and how they were implemented, with the detailed platform description be included in the supplementary, if possible, or in a url where this information could be reached.
Some vague statements should be better defined, e.g., react 'as quickly as possible'.

---

> ### Author Response · Authors · 2020-11-19
> **Response to Reviewer #1**
>
> Thank you for your comments and helpful suggestions. We address your specific questions and concerns below.
>
> **1. Presentation**
>
> We have revised the paper to include more discussions on the models and results, and moved the detailed system descriptions to the appendix.
>
> **2. Discussion of the model choices**
>
> We show the performance of variants of the goal inference module in the revised paper (first paragraph if 5.3). In particular, we test the performance of a model encoding the last observation from the demonstration. We also test taking the sequence of Alice’s ground-truth actions as input. The model using the last state obtains a precision and recall of 0.79 and 0.75, whereas the model with ground-truth actions obtains a precision and recall of 0.99 and 0.99 respectively. As a reference, the precision and recall of a random goal baselines is 0.08 and 0.09.
>
>
>
> **3. Ambiguous statement when stating “as quickly as possible”**
>
> Thank you for the suggestion, we have updated the paper to clarify this statement in the introduction. We describe “as quickly as possible” as taking the minimum number of parallel steps to complete each task.

---

### Author Response · Authors · 2020-11-19
**General Response**

We thank reviewers for their insightful feedback. We are encouraged that they find the challenge and benchmark a good resource to study aspects of Human-AI collaboration, and that they value the clarity and reproducibility of the paper. We include in this response the source code for the challenge and environment, which we will open source.

We have also revised the paper:
1. We added a more detailed discussion on the models and results and moved part of the description of the platform to the appendix.
2. We included more results on goal recognition for the *Watch* phase.
3. We included a more detailed discussion of the Help phase. In particular we tested the *Help* phase on tasks composed of multiple activities.
4. Fixed typos and included clarification notes.

The code can be downloaded in the [following link](https://osf.io/9rw3d/?view_only=a6507449ebe24d10aa55fb4f582761ce), downloading rebuttal_open_source.zip. The code contains a README with the instructions to test the simulator and run the baselines.

We provide more details in the direct responses to the reviewers.

---

### Decision · Program_Chairs · 2021-01-07
**Final Decision**

**Decision:**

Accept (Spotlight)

**Comment:**

Summary:
This paper provides an interesting and unique challenge problem on human-AI collaboration, with sample baselines. I think this is an extremely important topic and the community should embrace such challenge problems.

Discussion:
Reviewers agreed this paper should be accepted, particularly after seeing that ICLR has accepted such challenge papers in the past.

Recommendation:
I'd really like to see this get a spotlight as it would be great to highlight this innovative challenge to the community.